



# New methods to improve the vertical extrapolation of near-surface offshore wind speeds

Mike Optis[1], Nicola Bodini[1], Mithu Debnath[1], and Paula Doubrawa[1]

[1]National Renewable Energy Laboratory, Golden, Colorado, USA

**Correspondence:** Mike Optis (mike.optis@nrel.gov)

**Abstract.** Accurate characterization of the offshore wind resource has been hindered by a sparsity of wind speed observations that span offshore wind turbine rotor-swept heights. Although public availability of floating lidar data is increasing, most offshore wind speed observations continue to come from buoy-based and satellite-based near-surface measurements. The aim of this study is to develop and validate novel vertical extrapolation methods that can accurately estimate wind speed time series

across rotor-swept heights using these near-surface measurements. We contrast the conventional logarithmic profile against three novel approaches: a logarithmic profile with a long-term stability correction, a single-column model, and a machine-learning model. These models are developed and validated using 1 year of observations from two floating lidars deployed in U.S. Atlantic offshore wind energy areas. We find that the machine-learning model significantly outperforms all other models across all stability regimes, seasons, and times of day. Machine-learning model performance is considerably improved by

including the air-sea temperature difference, which provides some accounting for offshore atmospheric stability. Finally, we find no degradation in machine-learning model performance when tested 83 km from its training location, suggesting promising future applications in extrapolating 10-m wind speeds from spatially resolved satellite-based wind atlases.

*Copyright statement.* This work was authored by the National Renewable Energy Laboratory, operated by Alliance for Sustainable Energy, LLC, for the U.S. Department of Energy (DOE) under Contract No. DE-AC36-08GO28308. Funding provided by the Bureau of Ocean

Energy Management under Contract Number IAG-19-2122. The views expressed in the article do not necessarily represent the views of the DOE or the U.S. Government. The U.S. Government retains and the publisher, by accepting the article for publication, acknowledges that the U.S. Government retains a nonexclusive, paid-up, irrevocable, worldwide license to publish or reproduce the published form of this work, or allow others to do so, for U.S. Government purposes.

## 1 Introduction

The accurate characterization of the offshore wind resource is crucial for a range of analyses needed to support the growing offshore wind industry. Specifically, accurate time series estimates of wind speed across the rotor-swept heights of an offshore wind turbine are used for estimates of turbine and wind plant power production, which feed into various technical and economic analyses, ranging from grid integration (Mahoney et al., 2012), life-cycle cost analyses (Jong et al., 2017), and capacity expansion studies (Hasager et al., 2015).





Accurate characterization of rotor-swept offshore wind speeds has been hindered by the sparsity of observations at rotor-swept heights, especially in the U.S. offshore wind areas. Offshore meteorological towers are generally too expensive to install, especially up to 250 m–300 m, i.e., the expected upper rotor-swept heights of U.S. offshore wind turbines. Buoy-mounted floating lidar, however, are emerging as a game-changing technology, especially in the United States, providing accurate wind speed and direction measurements up to approximately 250 m (Carbon Trust, 2018); however, these units are also expensive,
mostly owned by wind plant developers, and their data are kept highly proprietary. In the United States, for example, as of December 2020, there are only six publicly available data sources for floating lidar in U.S. offshore waters (Table 1).

**Table 1.** Active Floating Lidar Deployments in U.S. Offshore Wind Energy Areas with Publicly Available Data (As of December 2020)

| Location | Time Resolution | Start Date for Public Data | Maximum Measurement Height | Data Access |
|---|---|---|---|---|
| Hudson South Call Area, New Jersey | 10 minute | 2019-09-04 | 200 m | DNV-GL (2020) |
| Hudson North Call Area, New Jersey | 10 minute | 2019-08-12 | 200 m | DNV-GL (2020) |
| Atlantic Shores, New Jersey | 10 minute | 2020-02-26 | 250 m | Atlantic Shores Offshore Wind (2020) |
| Mayflower, Massachusetts | Daily | 2020-04-13 | 250 m | Mayflower Offshore Wind (2020) |
| Humboldt, California | 1 second | 2020-10-01 | 250 m | Pacific Northwest National Laboratory (2020) |
| Morro Bay, California | 1 second | 2020-10-01 | 250 m | Pacific Northwest National Laboratory (2020) |

In place of rotor-swept height measurements, near-surface observations can be used as substitutes for characterizing the offshore wind resource (Mohandes and Rehman, 2018). The main data source is the network of buoy-based wind speed measurements from the National Data Buoy Center, maintained by the National Oceanic and Atmospheric Administration (National
Data Buoy Center, 1971). These data have been used to characterize the wind resource in offshore California (Wang et al., 2019; Optis et al., 2020c), the U.S. offshore Atlantic (Optis et al., 2020b), and the Great Lakes (Doubrawa et al., 2015). These buoys generally provide years worth of wind speed measurements less than 5 m and are of high quality. In addition to these buoys,





satellite-based scatterometer and synthetic-aperture radar measurements of the near-surface wind vector are increasingly being used to characterize the offshore wind resource (Doubrawa et al., 2015; Ahsbahs et al., 2017; Hasager et al., 2020; Ahsbahs

et al., 2020). These data are more spatially resolved than buoy-based wind speed data, but they are limited in their temporal coverage. Further, there is some error and uncertainty in how geophysical transfer functions are used to extrapolate the satellite measurements to the diagnosed 10-m wind speed that is disseminated (Kelly and Gryning, 2010; Badger et al., 2015).

This abundance of near-surface wind speed measurements is valuable for offshore wind resource characterization provided the measurements can be accurately extrapolated to rotor-swept heights. The conventional wind industry approach—the power

law profile—is not useful in this context because the method requires measurements at two heights to calculate the shear coefficient. The logarithmic wind profile (Monin and Obukhov, 1954), by contrast, is applicable and has a long history of accurately predicting wind speeds in the atmospheric surface layer (Holtslag, 1984; Troen and Petersen, 1989; Emeis, 2013); however, the logarithmic assumption has been shown to break down at rotor-swept heights under conditions of stable stratification as turbulent fluxes decrease in magnitude and near-surface winds begin to decouple from the winds aloft (Optis et al., 2014,

2016). Under such conditions, phenomena such as low-level jets can occur, which idealized models, such as the logarithmic wind profile—which assumes monotonically increasing wind speeds with height—are unable to account for.

Despite these shortcomings, the logarithmic profile still forms the backbone of the only extrapolation method that has been developed and validated for offshore applications. This method, developed by researchers at the Technical University of Denmark (DTU) in 2010, derives a stability-dependent long-term correction to the logarithmic wind profile (Kelly and

Gryning, 2010), where stability data (e.g., Obukhov length) are provided by numerical weather prediction simulations. This model (described in more detail in Section 3 and herein referred to as the DTU method) has been used in subsequent studies to extrapolate 10-m diagnosed winds from satellite products with good agreement with offshore observations in Europe (Badger et al., 2015; Hasager et al., 2020). The DTU method, however, can provide only a long-term mean wind profile extrapolation and is not useful when time series-based wind speeds across rotor-swept heights are needed (i.e., for most energy and economic

offshore wind analyses).

For such applications, two novel approaches with proven success on land but not thoroughly validated offshore could be suitable. The first is a single-column model (SCM) approach, in which a typical three-dimensional numerical weather prediction model is reduced to a single vertical dimension by assuming horizontal homogeneity (Baas et al., 2010). Further assumptions (described in Section 3) reduce the model to a simple set of differential equations that can be run efficiently on a personal

computer. The key advantage of the SCM is its ability to be forced at the lower boundary by wind and temperature observations. The SCM was used in Optis and Monahan (2016) and Optis and Monahan (2017) to extrapolate 10-m wind speeds up to 200 m at the Cabauw meteorological tower in the Netherlands. Results showed that the SCM performed about the same as the Weather Research and Forecasting (WRF) model (Skamarock et al., 2019) during a 10-year period, highlighting the benefit of local observations driving a highly simplified model.

Recently, machine learning has also emerged as a promising approach for the vertical extrapolation of wind speeds. Bodini and Optis (2020a) and Bodini and Optis (2020b) explored this concept using four lidars and surface flux stations dispersed around the Southern Great Plains site, operated by Argonne National Laboratory. They found that a relatively simple random





forest algorithm, trained on near-surface atmospheric variables, considerably outperformed the conventional power law and logarithmic wind profiles. This performance held even when a model was trained at one measurement site and tested at others up to 100 km away, i.e., through a round-robin approach. In the offshore environment, Vassallo et al. (2020) used a deep neural network to extrapolate near-surface winds in offshore California during a 1-month period, and they also found improvement relative to conventional techniques; however, the time period was short, and a round-robin approach was not applied.

The goal of this study is to assess the viability of these conventional and more novel extrapolation models for use in U.S. offshore areas. We provide comparisons among the different extrapolation models, and we benchmark against estimated wind profiles from the WRF model. We focus this study on the U.S. North Atlantic and Mid-Atlantic offshore areas, where the U.S. offshore wind industry is most developed (Musial et al., 2020). In Section 2, we describe the domain, the observations, and the WRF model setup used. Next, in Section 3, we describe the various extrapolation models. Intercomparisons of model performance are provided in Section 4, with concluding remarks provided in Section 5.

## 2 Data

### 2.1 Observations

To develop and validate the various extrapolation models, we leverage measurement data from two recently deployed floating lidars in offshore New Jersey and located within two current wind energy call areas (Figure 1). These lidars were deployed by the New York State Energy Research and Development Authority (NYSERDA), which has made data publicly available in real time through a web-based access portal (DNV-GL, 2020). The portal also includes detailed technical information regarding the lidars. An overview of these floating lidars and the data available are provided in Table 2. Lidar-measured wind speeds from 20 m to 200 m are used for the validation of the proposed extrapolation models (see Section 4), whereas the near-surface measurements at 2 m are used to develop and apply the extrapolation models (Section 3).

**Table 2.** Summary of Observational Data Set Being Analyzed

|  | Buoy E06 | Buoy E05 |
|---|---|---|
| Location | 39.55°N, 73.43°W | 39.97°N, 72.72°W |
| Period analyzed | Sep. 4, 2019–Aug. 16, 2020 | Aug. 12, 2019–Aug. 16, 2020 |
| Distance from coast | 69 km | 114 km |
| Lidar measurement heights | 20–200 m in 20-m increments | |
| Lidar variables | Wind speed, wind direction | |
| Surface variables | 2-m air temperature, sea surface temperature, 2-m wind speed, 2-m wind direction | |

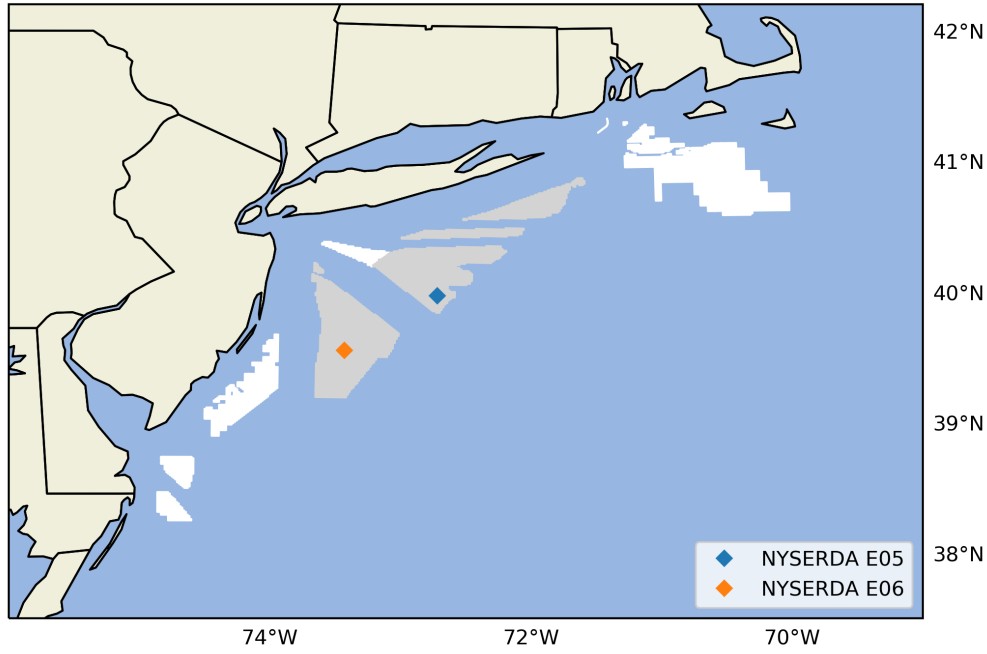

**Figure 1.** WRF simulation domain map considered in this study. The NYSERDA lidars are shown in blue and orange diamonds. White areas denote Bureau of Ocean Energy Management wind energy lease areas; gray areas denote Bureau of Ocean Energy Management call areas.

## 2.2 WRF Model

The WRF model is used in this study for two reasons. First, the DTU method (one of the extrapolation approaches considered in our analysis) requires surface atmospheric variables not available from the NYSERDA buoys. Second, validating the extrapolation models alongside WRF will provide key insights into the usefulness of novel extrapolation models for offshore wind energy and whether further development of these models is justified.

A summary of the WRF model setup is provided in Table 3, and the domain is shown in Figure 1. The WRF model is run from September 1, 2019, through August 31, 2020, in separate monthly runs. For each month, the simulation is initialized 2 days earlier (e.g., March 30 for April simulations) and run 1 day after the end of the month (e.g., May 1). The first day of the simulation is used to spin up the model from initial conditions, whereas the second and final days are used to stitch together the monthly runs into a single time series.

## 3 Extrapolation Models

In this section we describe the different wind speed extrapolation models considered in this study.



**Table 3.** Key Attributes of the WRF Model Used in This Study

| Feature | Specification |
| --- | --- |
| WRF version | 4.2.1 |
| Grid spacing for nested domains | 6 km, 2 km |
| Output time resolution | 5 minutes |
| Vertical levels | 61 |
| Near-surface-level heights (m) | 12, 34, 52, 69, 86, 107, 134, 165, 200 |
| Atmospheric forcing | ERA-5 reanalysis |
| Atmospheric nudging | Spectral nudging on 6-km domain, applied every 6 hours |
| Planetary boundary layer scheme | Mellor-Yamada-Nakanishi-Niino Level 2.5 |
| Microphysics | Ferrier |
| Longwave radiation | Rapid radiative transfer model |
| Shortwave radiation | Rapid radiative transfer model |
| Topographic database | Global multiresolution terrain elevation data from the U.S. Geological Survey and National Geospatial-Intelligence Agency |
| Land-use data | Moderate Resolution Imaging Spectroradiometer 30s |
| Cumulus parameterization | Kain-Fritsch |

### 3.1 Logarithmic profile

The logarithmic wind profile is given as:

$$U(z) = \frac{u_*}{\kappa} \left[ \ln\left(\frac{z}{z_0}\right) - \psi_m\left(\frac{z}{L}, \frac{z_0}{L}\right) \right] \qquad (1)$$

where $U$ is the wind speed, $\kappa$ is the von Kármán constant (typically taken to be 0.4), $z$ is the height above the surface, $u_*$ is the friction velocity, $z_0$ is the roughness length, $\psi_m$ is the stability function for momentum that adjusts the wind profile depending on atmospheric stability, and $L$ is the Monin-Obukhov length that characterizes surface layer atmospheric stability. The friction velocity, $u_*$, requires high-frequency sonic anemometer measurements that are not available at the NYSERDA buoys. To avoid specifying $u_*$, we reformulate Eq. 1 to use the 2-m buoy wind speeds as a reference measurement, allowing the wind profile to be calculated according to:

$$U(z) = U_{2m} \left[ \frac{\ln(z/z_0) - \psi_m(z/L, z_0/L)}{\ln(z_{ref}/z_0) - \psi_m(z_{2m}/L, z_0/L)} \right]. \qquad (2)$$



Here, we set $z_0$=0.0001 (which is the WRF output $z_0$ for offshore) and implement the $\psi_m$ formulations from Jiménez et al. (2012), which have become standard correction functions and are currently used in the WRF mesoscale model surface layer parameterization.

    The calculation of $L$ typically requires measurements of the momentum and turbulent temperature fluxes, which are not available from buoy measurements but require high-frequency three-dimensional wind speed components and temperature

measurements. Instead, we can calculate a "bulk" $L$ based on the bulk Richardson number, $Ri_B$:

$$Ri_B = \frac{g}{\theta_{avg}} \frac{z(\theta_z - \theta_{surf})}{U_z^2} \tag{3}$$

    where $z$ is the height 2 m above the surface, $g$ is the acceleration as a result of gravity, $\theta_{z_{2m}}$ is the potential temperature at 2 m, $\theta_{surf}$ is the potential temperature at the surface, and $U_{2m}$ is the 2-m wind speed. Combining Eq. 2 and Eq. 3 yields the following relationship between $L$ and $Ri_B$:

$$Ri_B = \frac{z}{L} \frac{ln\left(\frac{z}{z_0}\right) - \psi_h\left(\frac{z}{L}, \frac{z_0}{L}\right)}{\left[ln\left(\frac{z}{z_0}\right) - \psi_m\left(\frac{z}{L}, \frac{z_0}{L}\right)\right]^2} \tag{4}$$

    where $\psi_h$ is the stability function for temperature, also taken from Jiménez et al. (2012).

    Using Eq. 4, we iteratively solve for $L$ given $Ri_B$, which combined with Eq. 2 allows for the calculation of the vertical wind profile.

### 3.2   DTU Model

Noting the breakdown of the logarithmic wind profile in very stable conditions, the DTU method aims to preserve its applicability by applying it only in the context of a mean long-term wind profile, which is generally well estimated as logarithmic. The overall approach is to account for the distribution of $L$ values output from WRF throughout the year. As such, the DTU method is suitable only for long-term wind resource assessment because it requires at least 1 year of data and ideally many years (Kelly and Gryning, 2010).

The stability correction applied to the log extrapolation is height-dependent and computed based on empirical constants and atmospheric conditions at the site: the percentage of stable vs. unstable conditions; the quadratic mean of the kinematic heat flux; the mean, near-surface air temperature; and the time-averaged friction velocity. These input parameters are taken from the WRF simulations and are combined with stability functions, $\psi_m$, based on similarity theory to compute a vertical profile of the correction function (Figure 2). This correction is then added to the log extrapolation to yield a wind speed profile, as in

Eq. 1, where $u_*$ is taken from the WRF simulation, and $z_0$ is computed using the Charnock relationship, $z_0 = \alpha u_*^2/g$, with $g$ being the acceleration caused by gravity, and $\alpha = 0.0144$ (Charnock, 1955).

    First, we verify that the probability distribution functions for atmospheric stability are a good fit to the empirical distributions. This comparison is given in Figure 3. The functions shown in this figure take into account the percentage of stable vs.





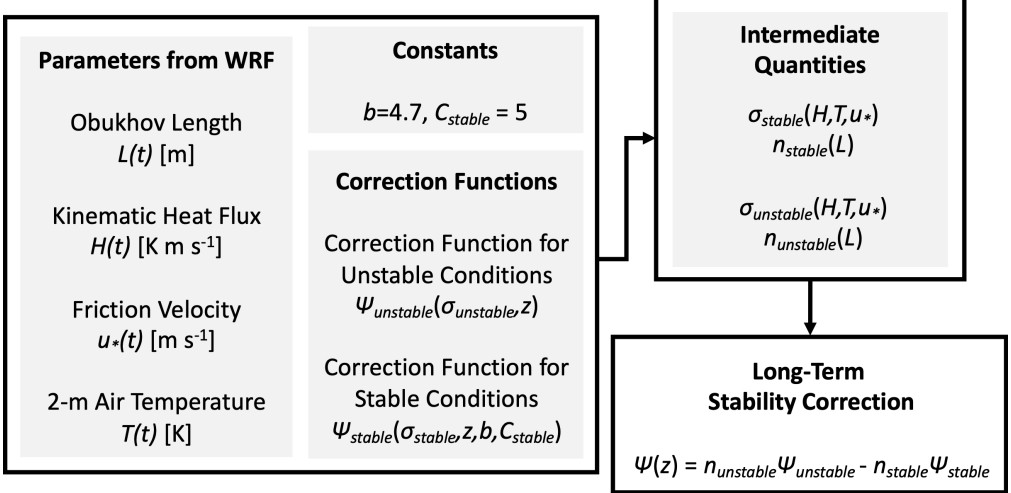

**Figure 2.** Schematic of quantities and calculations involved in the DTU model considered herein

unstable conditions at the NYSERDA buoy sites ($n_{stable}$ and $n_{unstable}$), scales of variation for $L^{-1}$ ($\sigma_{stable}$ and $\sigma_{unstable}$), and empirical constants ($C_{stable} = 5$ and $C_{unstable} = 12$). Note that previous work focusing on other data sets used different values for the $C\pm$ constants (e.g., both were set to 3.0 in Badger et al. (2015) to extrapolate satellite-derived wind speed measurements).

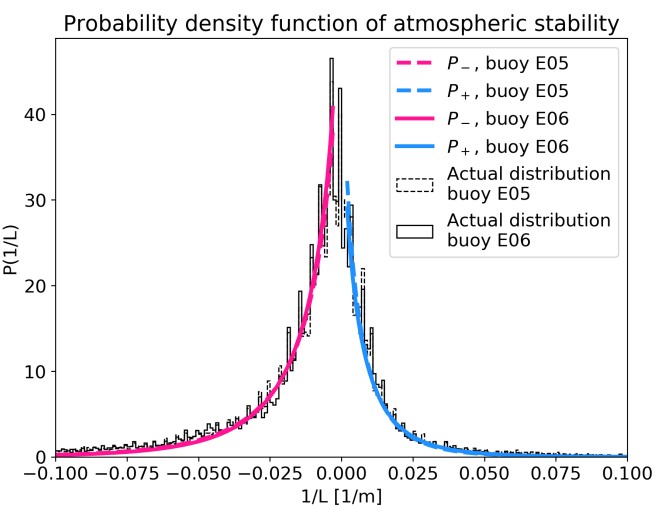

**Figure 3.** Empirical vs. theoretical distribution of atmospheric stability for the two buoy sites

The vertical profile of the stability correction function, $\psi_m$, obtained using the WRF and buoy measurements, is shown in Figure 4 for both buoys. The correction is unstable (i.e., positive) less than 80 m and stable (i.e., negative) greater than 80 m.





The two theoretical correction functions shown were calculated based on L=-500 m and L=500 m for the unstable and stable
   cases, respectively.

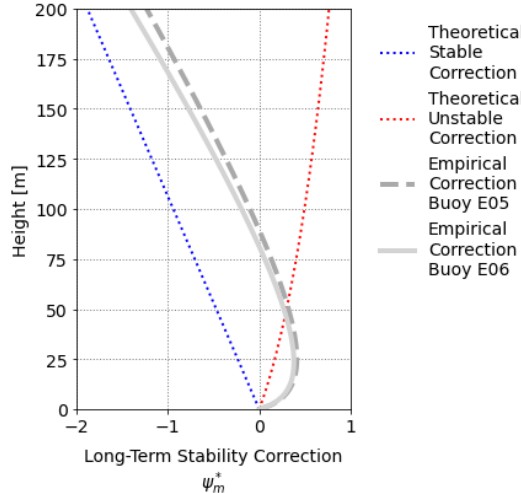

**Figure 4.** Empirical vs. theoretical stability correction profiles for the two buoy sites

### 3.3 Random Forest Machine-Learning Model

The third model considered is based on machine learning. Here, we consider a relatively simple ensemble-based regression
tree method, known as a random forest model, which has shown strong predictive power in previous land-based wind speed
extrapolation work (Bodini and Optis, 2020a, b) and in relating wind plant energy production to on-site atmospheric variables
(Optis and Perr-Sauer, 2019). We use the `RandomForestRegressor` module in Python's Scikit-learn (Pedregosa et al.,
2011). We consider a range of 10-minute averaged input variables available from the NYSERDA buoys: 2-m wind speed,
wind direction, pressure, and air temperature; the sea-surface temperature and air-sea temperature difference; as well as the
time of day and month of year. Wind direction, time of day, and month of year are all decomposed into their sine and cosine
components to preserve circularity (i.e., 0° and 360° directions are equivalent, as are 00:00 and 24:00).[1] A summary of these
variables is listed in Table 4.

   To ensure that the observation sets over which the random forest is trained and tested cover as much of the seasonal variability
as possible, we build the testing set using a consecutive 20% of the observations from each month in the period of record. We
evaluate different combinations of the hyperparameters with a fivefold cross-validation, and we randomly sample 20 sets. The
hyperparameters considered in the cross-validation and their sampled ranges are shown in Table 5. We evaluate the performance
of the learning algorithm based on the root mean square error (RMSE) between the measured and predicted wind speed

---

[1]Both are needed because each value of sine only (or cosine only) is linked to two different values of the cyclical feature.



**Table 4.** Input Features Used for the Random Forest Model

| Input Feature | Acronym | Measurement Height (m AGL) |
|---|---|---|
| 2-m wind speed | WS 2 m | 2 |
| Sine of 2-m wind direction | WD | 2 |
| Cosine of 2-m wind direction | | |
| 2-m air temperature | T | 2 |
| Sea-surface temperature | SST | 0 |
| Air-sea temperature difference | T - SST | - |
| 2-m air pressure | p | 2 |
| Sine of time of the day | Time | - |
| Cosine of time of the day | | |
| Sine of month | Month | - |
| Cosine of month | | |

at extrapolation height: the set of hyperparameters that leads to the lowest RMSE is selected and used to assess the final performance of the learning algorithm.

As described in detail in Bodini and Optis (2020b), it is both impractical and unfair to evaluate a machine-learning model
at the same site where it is trained. Critically, the model requires observations of the lidar-measured wind speeds up to 200 m to be trained. Evaluating model performance at the training site is impractical because the wind profiles are already known and unfair because the other extrapolation methods do not have such knowledge of lidar-measured wind profiles. Instead, model performance must be assessed through a round-robin approach, in which the model is evaluated at a site not used to train the model. Specifically, in this study, the random forest model is trained on data at NYSERDA buoy E05 and then
evaluated against other extrapolation models at NYSERDA buoy E06, located 83 km away, and then vice versa. This round-robin approach ensures a fair comparison of the different extrapolation methods and that no model has prior knowledge of lidar-measured wind profiles at the site where it is evaluated.

**Table 5.** Algorithm Hyperparameters Sampled in the Random Forest Cross-Validation

| Hyperparameter | Possible Values |
|---|---|
| Number of estimators | 10–800 |
| Maximum depth | 4–40 |
| Maximum number of features | 1–11 |
| Minimum number of samples to split | 2–11 |
| Minimum number of samples for a leaf | 1–15 |



## 3.4 Single-Column Model

The fourth model considered is an SCM. Essentially, it is a stripped-down version of a three-dimensional model, such as WRF,
in which only vertical exchanges are considered and horizontal homogeneity is assumed. This greatly simplifies the governing
equations of a three-dimensional model and reduces the SCM to a one-dimensional model in the vertical direction. By assuming
no moisture or cloud radiation, the equations of motion simplify further and depend only on the horizontal pressure gradients,
the Coriolis force, and the vertical turbulent flux of momentum and temperature:

$$\frac{\partial u}{\partial t} = f(v - v_G) - \frac{\partial(\overline{u'w'})}{\partial z}$$

$$\frac{\partial v}{\partial t} = f(u - u_G) - \frac{\partial(\overline{v'w'})}{\partial z}$$

$$\frac{\partial \theta}{\partial t} = \frac{\partial(\overline{\theta'w'})}{\partial z} \tag{5}$$

where $u$, $v$, and $w$ are the three vector wind components; $t$ is time; $z$ is the height above the surface; $\theta$ is potential temperature,
and $u_G$ and $v_G$ are the u- and v- components of the geostrophic wind. The $\overline{u'w'}$, $\overline{v'w'}$ terms represent the u- and v- components
of the vertical turbulent momentum flux, and $\overline{\theta'w'}$ represents the vertical turbulent temperature flux.

The momentum and temperature fluxes are not solved directly but rather parameterized based on well-established eddy-
diffusivity relationships:

$$\overline{u'w'} = -K_{\mathrm{m}}\frac{\partial u}{\partial z}$$

$$\overline{v'w'} = -K_{\mathrm{m}}\frac{\partial v}{\partial z}$$

$$\overline{\theta'w'} = -K_{\mathrm{h}}\frac{\partial \theta}{\partial z} \tag{6}$$

where $K_m$ and $K_h$ are the eddy diffusivities for momentum and temperature, respectively. These terms are themselves
parameterized with a range of possible options in the literature (Optis and Monahan, 2016, 2017). We adopt a relatively simple
first-order closure model that includes eddy diffusivities that are related to the wind speed gradient and a stability function that
depends on the Richardson number:





$$K_m = l_m^2 \frac{\partial U}{\partial z} f_m(R_i)$$

$$K_h = l_m l_h \frac{\partial U}{\partial z} f_h(R_i) \tag{7}$$

where $l_m$ and $l_h$ are the mixing lengths for momentum and temperature, respectively, and $f_m$ and $f_h$ are the stability

functions for momentum and temperature, respectively. There are a range of proposed formulations for the mixing lengths and stability functions. Here, we use the one developed by Smith (1990), which showed strong results when used in an SCM in previous studies (Optis and Monahan, 2016, 2017). A detailed explanation and the equations of the stability functions and mixing lengths can be found in Smith (1990); Cuxart et al. (2006); Optis and Monahan (2017).

The SCM equations are solved on a logarithmically stretched grid from a height of 2–2,000 m with 200 grid levels that

provide higher resolution near the surface. The lower boundary conditions at 2 m are the measured wind speed components and temperature from the NYSERDA buoys. The upper boundary conditions are the 800 hectopascal pressure-level data provided by the ERA-5 reanalysis. A zero-temperature gradient boundary condition is also applied at the top of the domain.

Recognizing that the geostrophic wind can change with height in conditions of horizontal temperature gradients, we calculate a geostrophic wind profile at each time step to force the simulations. This is done by first assuming that the 800-hPa winds from

ERA5 are geostrophic, which is a reasonable assumption at 2000 m, where surface friction effects should be negligible. Next, we calculate the geostrophic wind at the surface using surface pressure and air temperature data from the ERA5 reanalysis product:

$$u_G = -\frac{1}{f\rho} \frac{\partial P}{\partial y}$$
$$v_G = \frac{1}{f\rho} \frac{\partial P}{\partial x} \tag{8}$$

where $\rho$ is air density, and $P$ is pressure. The horizontal pressure gradient terms are calculated by taking a planar best fit of the closest nine ERA5 grid points that surround the buoy locations. Equation 8 is used to calculate the geostrophic wind components at 2 m, and finally the geostrophic wind profile is found by linearly interpolating the 2-m and 800-hPa values to the different SCM heights.

To initialize the simulation, we start by solving for the neutral vertical wind profile by imposing an equilibrium condition

(i.e., $\partial u/\partial t = 0$; $\partial v/\partial t = 0$; $\partial(\overline{\theta' w'})/\partial z = 0$). The simulation then moves forward from the neutral profile as a time-marching algorithm using the complete set of equations provided in this section. A continuous simulation is launched for the whole year of measurements without interruption.



## 4  Results

The four vertical extrapolation models presented in the previous section are all validated against lidar data from NYSERDA
buoys E05 and E06 during the full period of record. For each lidar, we consider only the time periods where wind speeds
are reported at every height from 20–200 m. Based on recent best-practice recommendations for validating offshore wind
models (Optis et al., 2020a), we validate the rotor-equivalent wind speed (REWS) rather than an assumed hub-height wind
speed. Details for calculating REWS are provided in (Wagner et al., 2014). To calculate REWS, we assume a 10-MW offshore
reference turbine as described in Beiter et al. (2020) and summarized in Table 6.

**Table 6.** 10-MW Offshore Reference Wind Turbine Specifications from Beiter et al. (2020) used to Calculate REWS

| Characteristic | Value |
| --- | --- |
| Rated power | 10 MW |
| Rotor diameter | 196 m |
| Hub height | 128 m |
| Rotor-swept heights | 30 m–226 m |

We also assess model performance using the four recommended performance metrics from Optis et al. (2020a), summarized
in Table 7. We note that the DTU method is capable of modeling only the mean wind profile; therefore, time series-based
performance analysis throughout this section excludes the DTU method.

**Table 7.** Performance Metrics Used to Assess Extrapolation Model Performance

| Name | Abbreviation | Description |
| --- | --- | --- |
| Bias | Bias | Difference between the mean modeled and observed result |
| Unbiased RMSE | cRMSE | The random error component after bias is removed, describing the differences in model variations around the mean |
| Square of correlation coefficient | $R^2$ | The correspondence or pattern between the modeled and observed variable |
| Earth-mover's distance | EMD | Difference between the probability distributions between the modeled and observed variable |



We begin with a comparison of the mean wind profile in Figure 5, showing results at both NYSERDA buoys E05 and E06. The observed wind profile shows moderate shear, increasing from approximately 8.5 m s$^{-1}$ to 10.5 m s$^{-1}$ at E05, and 8.0 m s$^{-1}$ to 10.3 m s$^{-1}$ at E06. As shown, the random forest machine-learning model provides excellent agreement with the mean profile, whereas the other models are deficient in some respects. The SCM underestimates wind speeds at E05 but is very close to the observed profile at E06. The logarithmic profile captures the upper winds relatively well with a slight positive bias, but it has increasingly higher bias at lower heights. The DTU method significantly overestimates wind speeds, especially at the upper heights, with nearly a 1.5-m s$^{-1}$ bias at 200 m. Finally, we see that the WRF model tends to underestimate the wind profile.

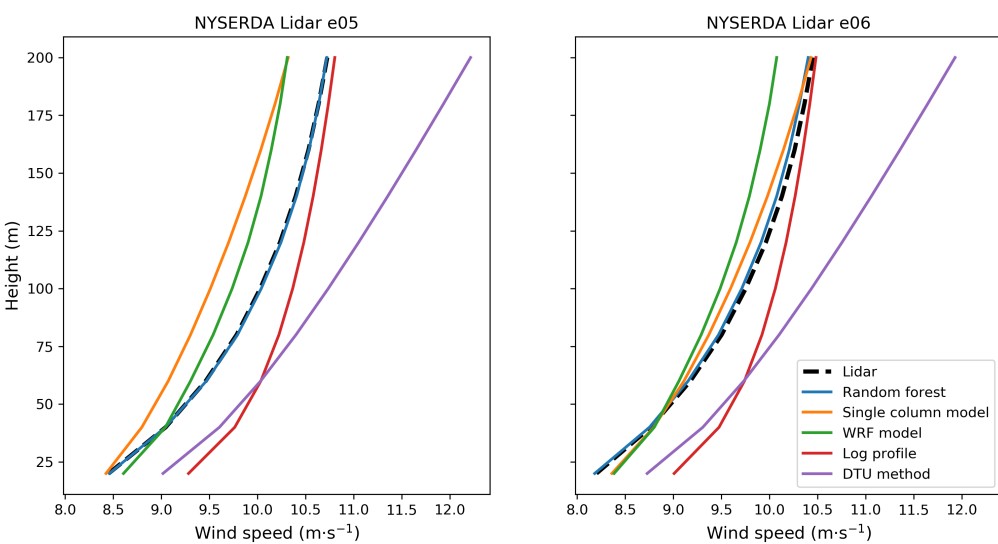

**Figure 5.** Mean modeled and observed wind profiles at NYSERDA buoys E05 and E06. The dotted line denotes the observed profile and solid colors denote the different extrapolation models.

REWS-based performance metrics for the different models are shown in Figure 6. Again, the strong performance of the machine-learning model is apparent, with considerably lower error metrics and higher correlation to observations relative to the other models. The bias is notably negligible at buoy E05 and slightly negative at E06. In contrast, the SCM has the weakest performance across all metrics at E05 and all but the bias at E06. The logarithmic profile performance falls in between the machine-learning model and the SCM and is the only model with a positive bias at both buoys. Finally, the WRF model tends to perform similarly to the logarithmic model, with slightly lower unbiased RMSE and higher correlation but higher magnitude of bias and earth-mover's distance (EMD).

Next, we consider the role of atmospheric stability in relative model performance. Here, we distinguish between unstable and stable conditions using the WRF-modeled bulk Richardson number, $Ri_B$, between 200 m and the surface ($Ri_B < 0$ for unstable conditions; $Ri_B > 0$ for stable conditions). Mean wind profiles by stability regime are shown in Figure 7. Here, we




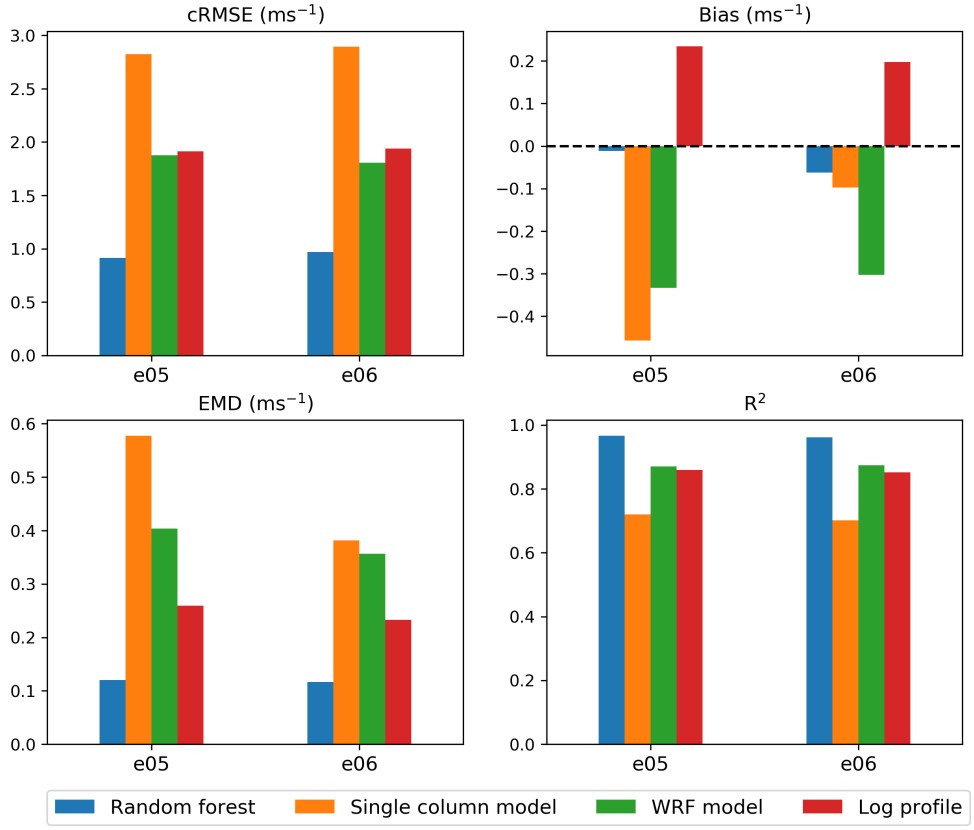

**Figure 6.** REWS performance metrics for the different vertical extrapolation models

focus only on buoy E05 and note that relative performance is similar at both buoys. The machine-learning model shows similar performance in unstable and stable conditions, accurately capturing the unstable profile and slightly underestimating the stable profile. The SCM performs reasonably well in unstable conditions but is unable to capture the high shear in the stable regime and significantly underestimates wind speeds. The log profile similarly underestimates wind speeds in stable conditions but overestimates in unstable conditions. Finally, the WRF model underestimates the wind profile in unstable conditions while accurately capturing winds greater than 100 m in stable conditions but overestimating them when less than 100 m. Overall, we see that all models apart from the random forest struggle with consistent accuracy across stability regimes.

This relative consistency is further illustrated in Figure 8, which shows the REWS performance metrics by stability regime. Again, we focus on buoy E05 and note the similar relative performance between models at buoy E06. We also see the random forest with the strongest performance metrics, apart from slightly higher magnitude bias and higher EMD in stable conditions relative to the WRF model. The SCM shows lower magnitude bias and EMD in unstable relative to stable conditions but high unbiased RMSE and correlation across both regimes. The log profile performs better in unstable conditions than stable





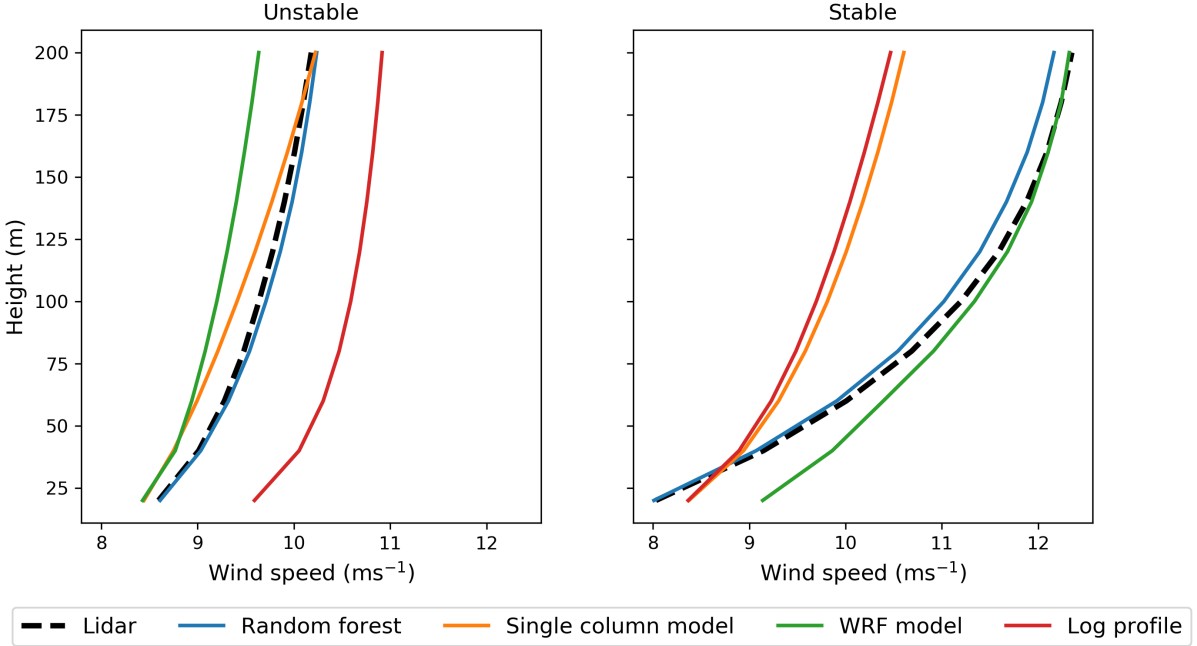

**Figure 7.** Mean modeled and observed wind profiles at NYSERDA buoy E05 in unstable (left) and stable (right) atmospheric conditions

conditions for all performance metrics, whereas the WRF model cRMSE and $R^2$ are lower in unstable conditions, but bias and EMD are higher relative to stable conditions.

Next, we present 12-by-24 heat maps to show the combined diurnal and monthly trends of model performance. We show only the bias heat maps in Figure 9, whereas the remaining performance metric heat maps are provided as supplementary material. We see that the machine-learning model has consistently low magnitude bias throughout the diurnal and monthly cycles, with no clear diurnal trends but a tendency to overestimate wind speeds in the fall. The SCM shows considerable negative bias throughout the year, with a tendency to overestimate wind speeds in November. Interestingly, the bias in December is positive from 01:00 to 12:00 and negative form 13:00 to 00:00. The WRF model shows some trends, with positive bias in spring in the early hours and negative bias in the middle hours. Finally, the logarithmic profile shows substantial trends, with strong overestimation of winds through most of the year and underestimation in spring, with the largest magnitude of the underestimates in the early hours.

### 4.1 Explaining DTU Model Performance

Figure 5 showed that the DTU method significantly overestimated wind speeds. This is a surprising result given its strong performance in Badger et al. 2015, in which 10-m satellite-measured winds were extrapolated. To explore this, we compare DTU model performance using both 2-m and 20-m measurements as the basis for extrapolation. The results are shown in Figure



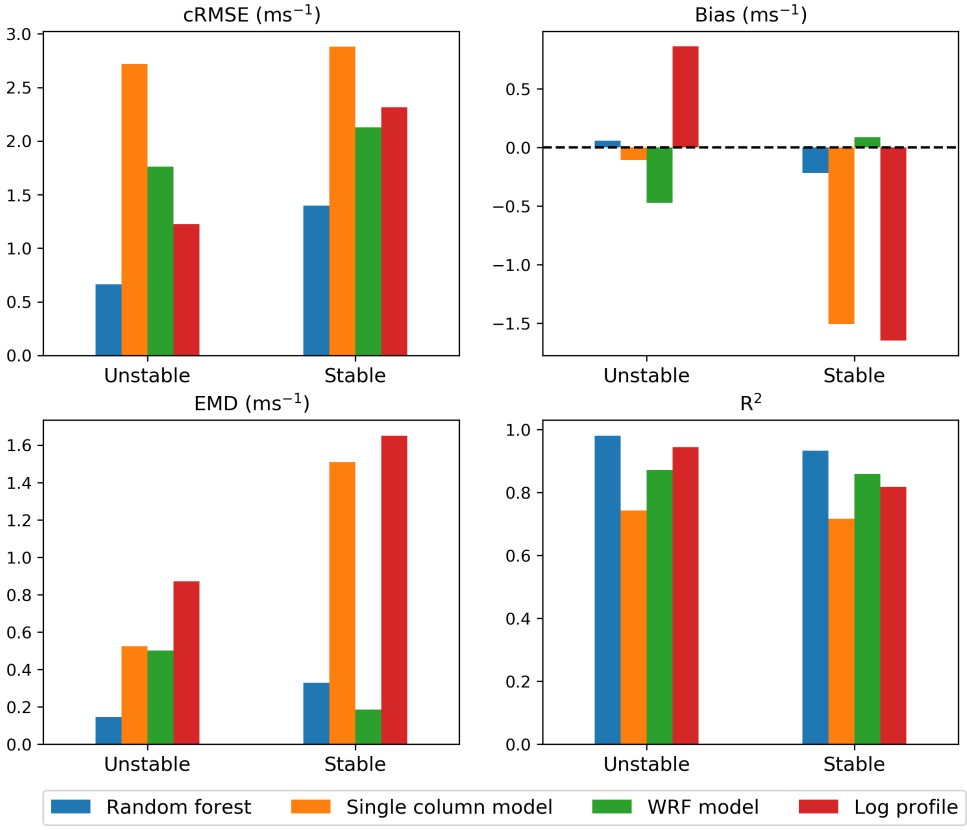

**Figure 8.** REWS performance metrics for the different vertical extrapolation models at NYSERDA buoy E05 for unstable and stable conditions

10. The extrapolation from the 2-m measurements does not match the measured wind speed profile. This is likely because the measurement height is too low and located within the viscous sublayer, where log-law approximations are not valid. When the same method is used to extrapolate from the 20-m lidar measurements, we see a good match between the extrapolated and measured values. This analysis reveals that the DTU method is not suitable for extrapolation based on buoy wind speed measurements, which are often made with propeller or cup anemometers between 2 m and 5 m above the sea surface. Instead, this method should be applied to short offshore meteorological masts and satellite-derived wind speed estimates.

### 4.2 Feature Importance in the Random Forest

Finally, we examine the random forest model in more detail given its strong performance in this study. Figure 11 shows the relative feature importance for each variable used to train the random forest model. Feature importance for the random forest model is calculated based on how many times the algorithm uses the variable to split the data, weighted by the improvement in







**Figure 9.** Heat maps (12 by 24) of REWS bias at NYSERDA buoy E05 for the different extrapolation models

model performance because of the split. Not surprisingly, the 2-m wind speed is the most important feature (nearly 80%). The second most important feature is the air-sea temperature difference at nearly 20%. This is an important result and highlights the influence of atmospheric stability on offshore wind profiles.

In fact, Debnath et al. (2020) found that a positive air-sea temperature difference was the key driver in the observed frequent occurrences of extreme wind shear and low-level jet events at the E05 and E06 buoys. Table 8 shows that including the air-sea temperature difference results in considerable improvements in random forest model performance, especially during the extreme high-shear cases identified in Debnath et al. (2020). Notably, the bias and EMD are both halved for the high-shear cases when using the air-sea temperature difference as an input feature.

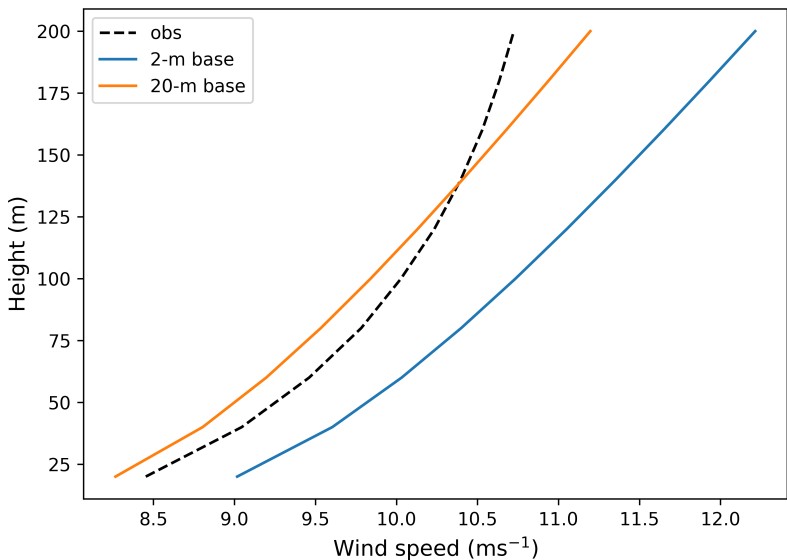

**Figure 10.** Mean observed and modeled wind profiles at NYSERDA buoy E05 when using the DTU method based on 2-m and 20-m measurements

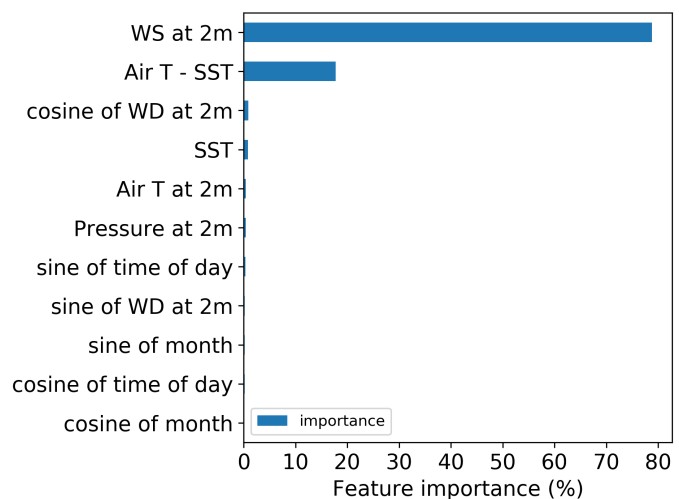

**Figure 11.** Relative feature importance for the random forest model in predicting 120-m wind speeds at NYSERDA buoy E05.

Finally, we examine how random forest model performance using the default round-robin approach (i.e., model trained and tested at different buoys) compares to that when trained and tested at the same site. In general, the model should perform best





**Table 8.** Performance Metrics at Buoy E05 for the Random Forest Model With and Without the Air-Sea Temperature Difference ($\Delta T_{air-sea}$) as an Input Feature

| Metric | All Data | | High Shear Cases | |
|---|---|---|---|---|
| | Without $\Delta T_{air-sea}$ | With $\Delta T_{air-sea}$ | Without $\Delta T_{air-sea}$ | With $\Delta T_{air-sea}$ |
| Bias (ms$^{-1}$) | 0.03 | 0.04 | -1.05 | -0.58 |
| cRMSE (ms$^{-1}$) | 1.07 | 0.84 | 1.46 | 1.29 |
| EMD (ms$^{-1}$) | 0.19 | 0.12 | 1.05 | 0.58 |
| R$^2$ | 0.95 | 0.97 | 0.89 | 0.91 |

when tested at the training site, as was found in Bodini and Optis (2020b). The degree of model deterioration with distance can provide insight into how well the model can generalize across space to perform extrapolation. The results of this comparison are shown in Table 9. Interestingly, at each site and for each metric, the round-robin performance is slightly better than the same-site performance. Accounting for the fact that the limited 1-year analysis contributes to some uncertainty in these metrics, it is clear that there is at best negligible model degradation throughout an offshore distance of 83 km. In contrast, Bodini and Optis (2020b) found that, on land, model performance decreased with distance from the training site, ranging from 11%–14% reductions throughout distances ranging between 40 km–100 km. The negligible performance reduction offshore—which can be attributed to the horizontal homogeneity of the offshore environment—has important implications for the applicability of machine-learning extrapolation techniques for all U.S. offshore waters using only a handful of lidar training sites.

**Table 9.** Comparison of Random Forest Model Performance when Trained and Tested Under a Round-Robin vs. a Same-Site Approach

| Metric | Buoy E05 | | Buoy E06 | |
|---|---|---|---|---|
| | Round Robin | Same Site | Round Robin | Same Site |
| Bias (ms$^{-1}$) | 0.07 | -0.09 | -0.05 | -0.02 |
| cRMSE (ms$^{-1}$) | 0.86 | 0.94 | 0.89 | 0.94 |
| EMD (ms$^{-1}$) | 0.13 | 0.16 | 0.09 | 0.13 |
| R$^2$ | 0.97 | 0.96 | 0.96 | 0.96 |

## 5   Conclusions

In this study, we developed novel methods for the vertical extrapolation of near-surface offshore wind speeds. We evaluated these methods against conventional extrapolation methods and WRF-modeled wind speeds using two floating lidars deployed in U.S. Atlantic wind energy call areas during a 1-year period. Of the four wind speed vertical extrapolation models considered, the random forest machine-learning model significantly outperformed the other models and accurately represented winds across the vertical profile in different seasons and times of day and in different stability regimes. Further, the random forest





model substantially outperformed the WRF model, highlighting the benefit of local observations in generating wind profiles. Moreover, the random forest model showed negligible to no performance decrease throughout the 83-km distance between the two floating lidars.

The SCM performance offshore could be improved considerably through better accounting of near-surface stability. The model was forced at its lower boundary only by the 2-m wind speed and temperature and critically did not consider the role of sea-surface temperature and related heat flux; therefore, the SCM really had no way to account for or to characterize the role of atmospheric stability, which was demonstrated in this study to be an important driver of the wind profile. In contrast, the WRF model can capture these effects, and the machine-learning model used the air-sea temperature difference, a proxy for

atmospheric stability, as an input variable, which considerably improved model results. Improving the SCM design to account for atmospheric stability (e.g., by substituting the temperature lower boundary condition with a flux-based measurement) should be an area of future work.

Results from this study clearly show the promise of a machine-learning-based approach to offshore wind extrapolation. It seems likely that models trained on only a handful of lidars dispersed in offshore waters could be sufficient to accurately

extrapolate wind speeds at all offshore locations in the surrounding area where surface measurements exist. This hypothesis should be tested more thoroughly using the additional floating lidars recently deployed in U.S. waters (Table 1). The ability for a machine-learning model to generalize across different oceans in particular (e.g., training a model in the Atlantic and testing it in the Pacific) would be an important area of future work as the U.S. offshore wind industry looks to Hawaii, the Pacific Northwest, and the Great Lakes for future expansion (Musial et al., 2020).

Applying the machine-learning approach to satellite-based wind speed observations would be the next future area of study. A collaboration between the National Renewable Energy Laboratory and DTU resulted in a U.S. Atlantic wind atlas at 10 m above sea level (Ahsbahs et al., 2020). Training and evaluating a machine-learning model at floating lidar sites using only data available across all the U.S. Atlantic area (i.e., satellite-measured winds and sea-surface temperature) would provide key insights into whether the Ahsbahs et al. (2020) wind atlas could be accurately extrapolated across offshore wind turbine

rotor-swept heights.

This proposed scope of future research will be aided by continued efforts to make floating lidar data public. Most deployed lidars are currently owned by wind energy developers and not publicly available. Public access to these data would greatly improve our understanding of the U.S. offshore wind resource and help produce more accurate hub-height observation-based offshore wind atlases.

*Code and data availability.* Observational data from the floating lidars is publicly available at DNV-GL (2020). The open-source WRF model was used for the numerical weather prediction simulations.



*Author contributions.*  MO wrote the manuscript, conducted the WRF simulations, and performed the inter-model comparison. NB built the random forest model and wrote Section 3.3, MD built the SCM model and wrote Section 3.4, and PD built the DTU model and wrote Section 3.2.

*Competing interests.*  The authors declare that they have no conflicts of interest.

*Acknowledgements.*  This work was supported and funded by the Bureau of Ocean Energy Management. We would like to thank Angel McCoy specifically for her support and guidance throughout this work. We also thank NYSERDA and DNV-GL for making the two floating lidar data publicly available, without which this study would not have been possible.



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
