# Peer review of "New methods to improve the vertical extrapolation of near-surface offshore wind speeds"

_Wind Energy Science, 2021_

## Referee Comment (RC1)

The study discusses vertical extrapolation methods for the estimation of wind speed time series from near-surface measurements. For that, the classical logarithmic approach has been compared to i) a single column model, ii) a logarithmic profile with a correction for long-term stability and iii) a machine learning approach using the Random Forest Regression. The authors could show that the machine learning approach is a valuable tool for vertical offshore wind extrapolation and discusses in addition the importance of the used features.

The manuscript is well-structured and the topic is interesting and of high relevance. The introduction describes the problem and state-of the-art, methods are well explained and results are presented in a clear way. Therefore, I would recommend accepting it with minor revisions in case the fact does not matter that most of the text and figures, except for the feature importance part, has already been published in the project report Optis (2020a)[1].

Specific comments:

Page 2, line 36/37: "These buoys generally provide years worth of wind speed measurements less than 5 m and" – please check this sentence, I guess the worth is not intended to be there. Also, I guess you mean at a height of less than 5 m?

Page 3, lines 50-51: This is really a beautiful, German sounding, nested sentence. I would suggest to ease it a bit.

Page 3, line 53: The separation between the classical and the corrected logarithmic profile is not clear here. As I understand, you talk about the classical approach in line 52/53 and afterwards about the corrected one developed by DTU, is that right? The confusing part is here the beginning of the sentence "This method, developed by researchers". It sounds to me as if you are referring to the classical method mentioned in the sentence before.

Page 3, line 61: I was expecting a description of the second novel approach somewhere below but could not clearly find one. Did you mean the machine learning approach? In that case, I would suggest to clearly state clearly that this is the second approach.

Page 9, line 164: please specify what you mean with "we randomly sampled 20 sets". Does it refer to the hyperparameter tuning?
* * *
[1] Optis, M., Bodini, N., Debnath, M., and Doubrawa, P.: Best Practices for the Validation of U.S. OffshoreWind Resource Models, Tech. Rep. NREL/TP-5000-XXXXX, National Renewable Energy Laboratory (NREL), Golden, CO (United States), 2020a.

---

## Author Comment (AC1)

**Response to Reviewer 2 – WES-2021-5**
**March 23, 2021**

We thank the reviewer for taking time to assess this manuscript and for the useful comments below. Please find our responses below in blue.

General Comments

The article considers improved characterization of offshore wind resource observations. The study is comparing the conventional logarithmic profile method against three novel approaches; a long-term stability correction, a single-column model, and a machine learning methodology. The result shows very promising results and that the machine-learning model significantly outperforms all other models.

The article is well written and structured. It does describe the three methodologies used well and include a good discussion of the result ending up in a conclusion.

We thank the reviewer for the positive feedback.

The article could benefit from an extended introduction to what is novel in the work presented. Three "novel" approaches are presented but it is not very clear what is new contributions and what is existing novel methods that are used for comparison. Perhaps the method name "DTU model" created this confusion. A short introduction to chapter 3 might be a suitable place to add it.

We agree with the reviewer that these novel methods could have been more clearly identified. As suggested, we have included this introductory paragraph in Section 3 to provide that clarity:

*"In this section we describe the different wind speed extrapolation models considered in this study. We first describe the conventional logarithmic wind profile and this discuss the DTU method, which is adopted for this study. We then discuss the most novel approaches that we have developed explicitly for this study, namely the single-column-model and machine learning methods."*

The discussion about the result in figure 4 is rather short and could benefit from increased clarity. It is perhaps also not needed information depending on previous question? The extended introduction with clarity on new contributions will probably solve this.

We agree with the reviewer that this figure was not adequately described. Given how we are merely adopting the DTU method, that it is not our novel contribution, and that the DTU

method did not perform nor is the focus of this study, we have decided to omit this figure from the manuscript and any related discussion.

A comment and discussion of the accuracy of the data used for comparison would also be suitable.

We have included in Section 2 that the lidar-measured wind speed uncertainty is 3.3%, as referenced from a recent energy assessment report published by DNV-GL.

---

## Author Comment (AC2)

**Response to Reviewer 1 – WES-2021-5**
**March 23, 2021**

We thank the reviewer for taking time to assess this manuscript and for the useful comments below. Please find our responses below in blue.

General Comments

The study discusses vertical extrapolation methods for the estimation of wind speed time series from near-surface measurements. For that, the classical logarithmic approach has been compared to i) a single column model, ii) a logarithmic profile with a correction for long-term stability and iii) a machine learning approach using the Random Forest Regression. The authors could show that the machine learning approach is a valuable tool for vertical offshore wind extrapolation and discusses in addition the importance of the used features.

The manuscript is well-structured and the topic is interesting and of high relevance. The introduction describes the problem and state-of the-art, methods are well explained and results are presented in a clear way. Therefore, I would recommend accepting it with minor revisions in case the fact does not matter that most of the text and figures, except for the feature importance part, has already been published in the project report Optis (2020a)[1].

We thank the reviewer for the general feedback and are happy to hear the topic and results are of relevance. We acknowledge that much of this work was recently published in the mentioned technical report, as was required as part of our contract with the Bureau of Ocean Energy Management. However, these technical reports generally do not reach a wide audience. This work in particular would reach even less given that it is hidden in the 2nd chapter of a report entitled "Best Practices for the Validation of Offshore Wind Resource Models." It is for these reasons that we felt it justified to publish in a wind energy journal that would attract a wider audience.

Ultimately, we defer to the editor on a resolution regarding the previously published technical report.

Specific Comments

Page 2, line 36/37: "These buoys generally provide years worth of wind speed measurements less than 5 m and" _–_please check this sentence, I guess the worth is not intended to be there. Also, I guess you mean at a height of less than 5 m?

Nice catch. We have corrected this sentence but have left the word 'worth' as this was added by NREL's communication team.

Page 3, lines 50-51: This is really a beautiful, German sounding, nested sentence. I would suggest to ease it a bit.

Hah, we're glad you liked the sentence, but we agree it's a mouthful. We removed the part about the monotonic increase of wind speed with height, which should be obvious to anyone with any familiarity with the subject.

Page 3, line 53: The separation between the classical and the corrected logarithmic profile is not clear here. As I understand, you talk about the classical approach in line 52/53 and afterwards about the corrected one developed by DTU, is that right? The confusing part is here the beginning of the sentence "This method, developed by researchers". It sounds to me as if you are referring to the classical method mentioned in the sentence before.

We have added "novel" in the first and second sentences in this paragraph to make clear the DTU method is the novel approach that builds on the logarithmic profile.

Page 3, line 61: I was expecting a description of the second novel approach somewhere below but could not clearly find one. Did you mean the machine learning approach? In that case, I would suggest to clearly state clearly that this is the second approach.

Yes the 2nd approach is the machine learning approach. We have modified to intro sentence to this approach to clarify this connection.

Page 9, line 164: please specify what you mean with "we randomly sampled 20 sets". Does it refer to the hyperparameter tuning?

Yes it refers to the hyperparameter turning. We have added to this sentence to make that point clear.